# The association between transition from metabolically healthy obesity to metabolic syndrome, and incidence of cardiovascular disease: Tehran lipid and glucose study

**Farhad Hosseinpanah** [1☯‡*], **Erfan Tasdighi** [1☯‡], **Maryam Barzin** [1], **Maryam Mahdavi** [1], **Arash Ghanbarian** [2], **Majid Valizadeh** [1], **Fereidoun Azizi** [3]

**1** Obesity Research Center, Research Institute for Endocrine Sciences, Shahid Beheshti University of Medical Science, Tehran, Iran, **2** Prevention of Metabolic Disorders Research Center, Research Institute for Endocrine Sciences, Shahid Beheshti University of Medical Sciences, Tehran, Iran, **3** Endocrine Research Center, Research Institute for Endocrine Sciences, Shahid Beheshti University of Medical Science, Tehran, Iran

☯ These authors contributed equally to this work.
‡ These authors are joint senior authors on this work.
* fhospanah@endocrine.ac.ir

**Data Availability Statement:** The data set is the property of Research Institute for Endocrine

## Abstract

Considering that the data available on the cardiovascular (CV) risk of metabolically healthy obesity phenotype, and the effect of transition to an unhealthy status are inconsistent, the aim of this study was to investigate the possible role of transition to unhealthy status among metabolically healthy overweight/obese (MHO) subjects on CVD incidence over a median follow-up of 15.9 years. In this large population-based cohort, 6758 participants (41.6% men) aged ≥ 20 years, were enrolled. Participants were divided into 4 groups based on their obesity phenotypes and follow-up results, including persistent metabolically healthy normal weight (MHNW), persistent MHO, transitional MHO and metabolically unhealthy overweight/obese (MUO). Metabolic health was defined as not having metabolic syndrome based on the Joint Interim Statement (JIS) criteria. Multivariable adjusted hazard ratios (HRs) were calculated for cardiovascular events. During follow-up, rate of CVD Incidence per 1000 person-years were 12 and 7 in males and females, respectively. Multivariable adjusted HRs (CI 95%) of CVD incidence among males and females were 1.37 (.78–2.41) and .85 (.34–2.15) in persistent MHO group, 1.55 (1.02–2.37) and .93 (.41–2.12) in transitional MHO group and 2.64 (1.89–3.70) and 2.65 (1.24–5.68) in MUO group. Our findings showed that CVD risk did not increase in the persistent MHO phenotype over a 15.9-year follow-up in both sexes. However, transition from MHO to MUO status during follow-up increased the CVD risk just in male individuals. Further studies are needed to provide conclusive evidence in favor of benign nature of transitional MHO phenotype in females.

Sciences (RIES) and is made available upon
approval of the research proposal by the research
council and the ethics committee. The RIES ethics
committee must issue an approval in case of a
request for access to the de-identified dataset. Data
request may be sent to the head of the RIES Ethics
Committee, Dr. Azita Zadeh-Vakili, at email:
azitavakili@endocrine.ac.ir.

**Funding:** The authors received no specific funding
for this work.

**Competing interests:** The authors have declared
that no competing interests exist.

## Introduction

Obesity is a notorious risk factor for cardiovascular disease (CVD) and its prevalence continues to rise rapidly, throughout the world [1]. Similarly, prevalence of obesity (BMI$\geq$ 30 kg/m$^2$), which is reported to be 17.4%, has an ascending trend in Iranian population [2]. Variable distribution of metabolic risk factors across the spectrum of BMI has resulted in different obesity phenotypes. metabolically healthy overweight/obese (MHO) is a subgroup of individuals which does not accompany typical obesity associated metabolic disorders; however, a precise definition is still not defined [3, 4]. MHO prevalence is a matter of debate but it has been reported to be 6% to 75% in various populations and based on different definitions [5].

MHO phenotype is considered as a dynamic or transient phenotype, since nearly half the subjects lose their metabolic health during a 10 year follow-up [6, 7]. This transitional feature of MHO could cause a heterogeneity which divides this phenotype into two subgroups: persistent healthy and transitional. Recently, the prognostic value of MHO has become a challenging subject. While a few studies suggest that MHO is a benign phenotype [6, 8], longer prospective studies showed that, the risk of CVD this phenotype is between the normal weight healthy status and metabolically unhealthy [9–11]. This inconsistency could be explained by different lengths of follow-up, as long term studies are more likely to detect the transitional subgroup of MHO [12]. Moreover, the definition of metabolically unhealthy status can be another reason for this inconsistency, as studies with a more strict definition of metabolic state found no CVD risk in the MHO phenotype [13].

Few studies have investigated CVD risk in the transitional subgroup of MHO compared to the persistent healthy [14, 15]; therefore, in this prospective cohort study, we assessed CVD outcomes in MHO subjects who became unhealthy during a 15.9 year Follow up, separately in males and females.

## Materials and methods

### Study population

The Tehran Lipid and Glucose Study (TLGS) is an ongoing prospective population-based study, conducted to determine the risk factors for non-communicable diseases among a representative Tehranian urban population [16]. In the TLGS, 15,005 Participants, aged over 3 years, were selected by a multistage cluster random sampling method. A questionnaire for past medical history and data was completed during interviews; blood pressure and anthropometrical measurements and a limited physical examination were performed and lipid profiles, fasting blood sugar and 2-hours-postload-glucose challenge were measured. Rose angina questionnaire is completed for individuals over 30 years of age. Details of the study protocol are available elsewhere [16]. At the beginning of the study, all participants provided a written informed consent, and the study was approved by the research institute Endocrine Science ethics committee and was conducted in accordance with the principles of the Declaration of Helsinki. For the current study, 12808 participants were recruited during the first (1999–2001) or second phase (2002–2005) of TLGS. Based on the original TLGS study protocol, those with chronic and debilitating conditions at baseline (e.g. chronic renal or hepatic disease, etc.) were not recruited. After exclusion of those who were aged<20 years (n = 186), were pregnant (n = 97), had cancer (n = 53), had history of cardiovascular disease (CVD) at baseline (n = 579), chronic use of corticosteroids (n = 246), those with BMI<18.5 kg/m2 (n = 293), those with metabolically unhealthy normal weight status (n = 510), those who had missing values for anthropometric or metabolic data (n = 683) and those with missing CVD data at baseline (n = 808), 9353 participants were selected for categorization of phenotype, and analyses of

the follow-up data, until 2017 with median follow-up of 15.9 years (11.6–16.4). Lost to follow-up rate was 7% (n = 685). Of these participants, 1910 (20.4%) subjects with non-persistent MHNW status or MHO participants who revert to normal weight status were excluded, and the final analysis were performed on 6758 participants with complete data (Fig 1).

**Measurements.** Subjects were interviewed privately, by trained interviewers using pre-tested questionnaires. Initially, information on age, sex, education, medical history of CVD,

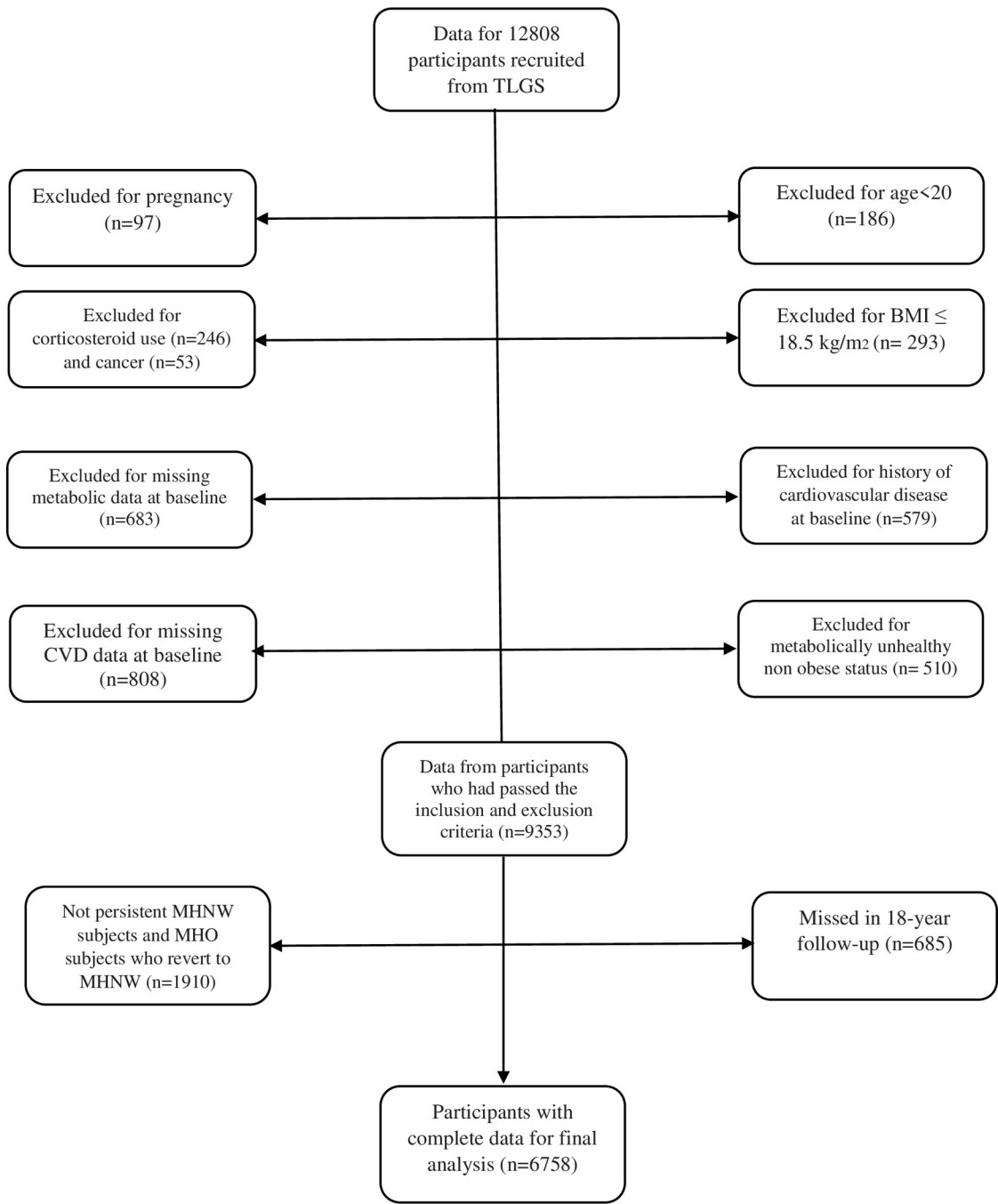

**Fig 1. Diagram showing the selection process of study participants.**

medication use, smoking habit, physical examination results, and family history of diabetes and premature coronary heart disease, was collected.

Weight was measured to the nearest 100 g while participants were minimally clothed and barefoot, using digital scales. Height was measured using a tape meter, while participants were in standing position and barefoot, with shoulders in normal alignment. BMI was calculated as weight in kilograms divided by height in meters squared. WC was measured at the level of the umbilicus using an un-stretched tape meter, without any pressure to the body surface, and recorded to the nearest 0.1 cm. All measurements were taken by the same person. To measure blood pressure, participants were first asked to rest for 15 min, then a qualified physician took the systolic blood pressure (SBP) and diastolic blood pressure (DBP) twice in a seated position, after one initial measurement for determining the peak inflation level, using a standard mercury sphygmomanometer. The mean of the two measurements was considered as the participant's blood pressure.

Blood samples were drawn from all the study participants after 12–14 hour of overnight fasting, and all analyses were undertaken at the TLGS research laboratory on the day of blood collection, using selectra 2 auto-analyzer (Vital Scientific, Spankeren, the Netherlands). Fasting blood sugar (FBS) was measured by the enzymatic colorimetric method using glucose oxidase. For lipid measurements, total cholesterol (TC) and triglyceride (TG) levels were assayed by relevant kits (Pars Azmoun, Tehran, Iran) using enzymatic colorimetric tests with cholesterol esterase and cholesterol oxidase, and glycerol phosphate oxidase, respectively. High density lipoprotein cholesterol (HDL-C) was measured with phosphotungstic acid. All samples were analyzed when internal quality control met the acceptable criteria. Inter- and intra-assay coefficients of variations at baseline were 2.2% for serum glucose, 2.0% and 0.5% for HDL-C and 1.6% and 0.6% for TG, respectively. Details of all measurement methods are available elsewhere [16]. Regarding measurement of fasting serum insulin by electrochemiluminescence immunoassay (ECLIA), Roche Diagnostics kits and the Roche/Hitachi Cobas e-411 analyzer (GmbH, Mannheim, Germany) were used. Intra- and inter-assay coefficients of variation were 1.2 and 3.5%, respectively.

**Definition.** Metabolically unhealthy was defined using the criteria proposed by the Joint Interim Statement (JIS) [17] as follows: (1) FBS ≥100 mg/dl (5.6 mmol/l) or 2-h blood glucose ≥140 mg/dl (7.8 mmol/l) or drug treatment; (2) fasting TGs ≥150 mg/dl (1.7 mmol/l) or drug treatment; (3) fasting HDL-C <50 mg/dl (1.29 mmol/l) in women and <40 mg/dl (1.03 mmol/l) in men or drug treatment; (4) raised blood pressure defined as SBP≥ 130 mmHg, DBP≥ 85 mmHg or antihypertensive drug treatment; (5) WC ≥ 89/91 cm in men /women based on national cut-offs [18]. Metabolically healthy status was considered as having ≤2 of the JIS components, and participants with 3 or more criteria were considered metabolically unhealthy.

Data regarding serum insulin level was available only for 3946 subjects. Insulin resistance (IR) was calculated as follows: homeostatic model assessment-insulin resistance (HOMA-IR) = [fasting insulin (μU/mL) × fasting glucose (mmol/L)]/22.5. IR was defined as HOMA-IR ≥ 2.6 in both sex [19].

Details on data collection of CVD outcome have been published elsewhere [20]. Coronary heart disease (CHD) included cases of definite myocardial infarction (diagnostic electrocardiographic results and biomarkers), probable myocardial infarction (positive electrocardiographic findings plus cardiac symptoms or signs plus missing biomarkers or positive electrocardiographic findings plus equivocal biomarkers), proven CHD by angiography, and death due to CHD. CVD was defined as any CHD, stroke (a new neurological deficit that has lasted for24 h), or CVD death (fatal CHD or fatal stroke).

Family history of premature CAD was defined as previous diagnosis of CAD in first-degree female relatives aged <65 years or first-degree male relatives aged <55 years. Smoking status was defined as nonsmoker and smoker (ex-smoker, current or occasionally). Educational level was categorized based on years of education (12 years, >12 years). Physical activity was assessed by the Lipid Research Clinic (LRC) questionnaire in the first phase of TLGS. Due to the lack of precision of the LRC [21], the Modifiable Activity Questionnaire (MAQ), which measures all three types of activity (leisure time, job and household activities) [22], was used in the rest of follow-up examinations. Since the duration of physical activity was not accounted in the LRC, participants who were enrolled in the study from the first examination of TLGS, were considered to be physically active if participating in vigorous physical activity for a minimum of 3 days per week. Individuals who entered the study at the second follow-up examination of TLGS were defined as physically active if they achieved a minimum of at least 600 MET (metabolic equivalent task)-minutes per week [23].

Overweight/obesity was defined as BMI ≥ 25 kg/m2. According to BMI categories and metabolic status, participants were divided into 4 groups: (1) metabolically healthy normal weight (MHNW) defined as BMI<25kg/m2 and healthy metabolic status; (2) metabolically healthy overweight/obese (MHO) defined as BMI≥25kg/m2 and healthy metabolic status; (3) metabolically unhealthy normal weight (MUNW) defined as BMI<25kg/m2 and unhealthy metabolic status; (4) metabolically unhealthy overweight/obese (MUO) defined as BMI≥25kg/m2 and unhealthy metabolic status. Persistent MHNW group was defined as individuals with MHNW phenotype which did not change during follow-up. Transitional MHO subgroup was defined as MHO individuals who developed metabolic abnormalities at any time during follow-up. Persistent healthy group was defined as MHO individuals who stayed metabolically healthy throughout the follow-up.

## Statistical analysis

Normally-distributed and skewed continuous variables were illustrated as mean±SD and median (IQR 25–75), respectively. Categorical variables of baseline characteristics were shown as frequency (percentages). The baseline characteristics of all participants based on obesity phenotypes were compared. Statistical analysis for continuous and categorical variables was performed using One Way ANOVA and Chi-Square test, respectively. Post hoc analysis with bonferroni correction was applied for pairwise comparison between each group of obesity phenotypes.

To determine the association between obesity phenotype and the incidence of CVD events in metabolically healthy participants, those individuals whose obesity phenotype was not evaluated during follow-up or who had CVD before the assessments, were excluded. Transition from metabolically healthy obesity to metabolic syndrome was time-varying until incidence of cardiovascular disease or till the end of follow-up. This variable, due to the nature of its definition, had to change over time. Therefore, BMI and all metabolic criteria (FBS, 2hpG, TG, HDL-C), SBP, DBP and waist circumference were measure at baseline and at each phase of TLGS which were three years apart until the occurrence of the outcome or till the end of follow-up.

The association between obesity phenotype and incidence of CVD was analyzed using Cox proportional hazards. Hazard ratios (HRs) with 95% confidence intervals for each sex group were used to estimate the incidence of CVD events. The person-year, which was assessed to obtain CVD incidence rates, was reported as number of cases per 1000 person years. The analyzed factors were independent unadjusted factors (only obesity phenotype), and adjusted variables for age, smoking (non-smokers as reference), total cholesterol, physical activity (≥600

MET as reference), family history of CVD and educational levels (illiterate/primary as reference). The event date for the incident cases of CVD was defined from baseline phase of the study to the first incident CVD event and for those with negative event (censored subjects), the time was the interval between the first and the last observation dates. Significant interaction was found between sex and obesity phenotype with incidence of CVD events ($p$-value $< .001$). The last observation carried forward (LOCF) method was used to handle the missing data of phenotype status in every phase. All analyses were performed using SPSS software, version 20 (SPSS, Chicago, IL, USA) and Stata software, version 14.0 (Stata Corp LLC, TX, USA); the differences with a P-value greater than 0·05 were considered significant (two-tailed test).

## Results

This study included 6758 (2812 males) individuals with mean age of 43.5 ± 14.5 and 42.3 ± 13.1years for males and females respectively. At baseline, 472 (16.8%) males and 411 (10.4%) females were considered MHNW, who had the same obesity phenotype throughout the follow-up. MUO counterparts at baseline were 1585 (56.4%) and 1978 (50.1%) males and females, respectively. In the male group, 755 (26.9%) participants had the MHO phenotype, of whom 66.6%, (n = 503) became metabolically unhealthy during follow-up. Female participants with MHO phenotype at baseline were 1557 (39.4%), of whom 57.3% (n = 893) became metabolically unhealthy during follow up.

Table 1 shows the baseline characteristics of participants according to different obesity phenotypes. Individuals with MUO phenotype at baseline were older, had higher BMI and a worse metabolic profile than other phenotypes. Persistent MHO females, unlike male participants, were significantly younger and had lower BMI than the transitional counterparts. Regarding metabolic parameters, including TG, TC, HDL-C and blood pressure, persistent MHO females had a better profile compared to transitional counterparts. Persistent MHO males had higher HDL and lower TG than transitional counterparts. In both sex groups, highest prevalence of Insulin resistance was in MUO individuals. MHO subjects in both sex groups had lower prevalence of Insulin resistance than MUO subjects, although it was higher than MHNW participants. Comparing females with males just in the transitional group, revealed that females had lower SBP, FBS and TG and higher HDL-C than their male counterparts. In contrast to male participants, there was a significant difference in females regarding education status and physical activity level; however, pairwise comparisons between persistent MHO and transitional MHO with the reference group (MHNW) did not reveal any difference in this regard. Moreover, between-group analysis showed that there were no differences between male and females in persistent and transitional MHO regarding education status and physical activity.

During a 15.9-year follow-up, 450 and 378 new CVD events occurred in males and females, respectively. Incidence rate per 1000 person-years was 12 and 7 in males and females, respectively. Fig 2 represents Kaplan-Meier curves for cumulative survival free from CVD as a function of obesity phenotypes, stratified by BMI and metabolic health. As shown, the survival curves differed significantly in both sex groups (log rank test, p < 0.001).

HRs for incident CVD in different obesity phenotypes among 6758 study participants are shown in Table 2. HRs for CVD incidence in MUO participants, according to the fully adjusted model, were 2.64 (1.89–3.70) and 2.65 (1.24–5.68) in males and females, respectively. In all models, persistent MHO individuals (both sex groups) did not have a significant risk for CVD. The transitional phenotype did not have a significant risk for CVD in female subjects, based on all models [HR = 1.94 (.86–4.41), HR = 1.09 (.48–2.48), HR = .93 (.41–2.12)]. Male subjects with transitional phenotype did not have a significant risk for CVD in the unadjusted

**Table 1. Baseline characteristics of 6758 study participants according to obesity phenotypes (body mass index and metabolic health) and gender.**

| | Total | Persistent MHNW | Persistent MHO | Transition from MHO to MUO | MUO | *P-value* |
|---|---|---|---|---|---|---|
| **Male** | | | | | | |
| Number | 2812 | 472 | 252 | 503 | 1585 | - |
| Age (year) | 43.5 ± 14.5 | 42.35 ± 16.7 | 37.6 ± 13.0 | 39.9 ± 12.3 | 46.0 ± 14.5 | < .001 |
| Weight(kg) | 79.3 ± 12.7 | 61.7 ± 6.0 | 81.2 ± 9.5 | 80.1 ± 8.7[f] | 84.0 ± 11.0 | < .001 |
| BMI (kg/m$^2$) | 27.4 ± 3.9 | 21.2 ± 1.6 | 27.9 ± 2.5 | 27.7 ± 2.3[f] | 29.1 ± 3.0 | < .001 |
| WC (cm) | 93.4 ± 10.8 | 77.3 ± 6.0 | 92.9 ± 7.9 | 92.6 ± 7.6 | 98.6 ± 8.0 | < .001 |
| Smoker, n (%) | 726 (25.8) | 147 (31.2) | 56 (22.2) | 125 (24.9) | 398 (25.1) | .024 |
| Education, n (%) | | | | | | |
| Diploma and Less than diploma | 2301 (81.9) | 382 (80.9) | 193 (76.9) | 407 (80.9) | 1319 (83.3) | .07 |
| Higher than diploma | 508 (18.1) | 90 (19.1) | 58 (23.1) | 96 (19.1) | 264 (16.7) | .07 |
| Physical activity, n (%) | | | | | | |
| Low | 1973 (69.4) | 318 (67.8) | 168 (67.5) | 351 (70.1) | 1100 (69.9) | .72 |
| High | 855 (30.6) | 151 (32.2) | 81 (32.5) | 150 (29.9) | 473 (30.1) | .72 |
| Family history of premature CAD, n (%) | 427 (15.2) | 57 (12.1) | 41 (16.3) | 78 (15.5) | 251 (15.9) | .22 |
| SBP (mmHg) | 121.7 ± 1802 | 112.7 ± 16.4 | 113.7 ± 13.3 | 116.2 ± 12.8[f] | 127.4 ± 18.7 | < .001 |
| DBP (mmHg) | 78.8 ± 11.2 | 71.8 ± 10.2 | 74.0 ± 9.0 | 75.8 ± 8.7 | 82.6 ± 10.9 | < .001 |
| Hypertension, n (%) | 2166 (77.3) | 431 (91.5) | 234 (92.9) | 466 (92.6) | 1035 (65.7) | < .001 |
| FBS (mg/dl) | 98.7 ± 29.2 | 89.2 ± 15.4 | 88.7 ± 10.5 | 91.3 ± 14.4[f] | 105.5 ± 35.4 | < .001 |
| 2-hBG (mg/dl) | 117.2 ± 59.8 | 94.1 ± 29.2 | 96.0 ± 30.0 | 106.6 ± 42.9 | 131.5 ± 70.5 | < .001 |
| Diabetes, n (%) | 320 (11.7) | 13 (2.9) | 3 (1.2) | 20 (4.1) | 284 (18.4) | < .001 |
| HDL cholesterol(mg/dl) | 37.1 ± 8.9 | 42.5 ± 9.7 | 43.1 ± 8.3 [e] | 39.3 ± 8.6[f] | 33.8 ± 7.2 | < .001 |
| Low HDL cholesterol[c] (%) | 1927 (68.6) | 212 (44.9) | 93 (36.9) [e] | 267 (53.1)[f] | 1355 (85.7) | < .001 |
| Total cholestrol (mg/dl) | 205.1 ± 43.3 | 183.8 ± 41.7 | 195.1 ± 39.1 | 202.8 ± 41.5 | 213.7 ± 43.3 | < .001 |
| Triglycerides (mg/dl) | 169 (114–241) | 94 (72127) | 108 (84–136) [e] | 135 (109–179)[f] | 215 (169–291) | < .001 |
| Triglycerides ≥ 150 mg/dl | 1658 (59.0) | 83 (17.6) | 38 (15.1) | 170 (33.8) | 1367 (86.2) | < .001 |
| Insulin[d] (Mu/L) | 7.8 (5.5–10.70 | 4.7 (3.2–6.7) | 7.6 (5.5–9.7) | 7.1 (5.7–9.3) | 9.3 (6.8–12.6) | < .001 |
| HOMA-IR[d](mole ×mU/l$^2$) | 1.8 (1.2–2.6) | 1.0 (0.7–1.4) | 1.5 (1.1–2.1) | 1.6 (1.2–2.1) | 2.3 (1.5–3.0) | < .001 |
| IR, n (%) | 358 (26.2) | 2 (0.9) | 16 (13.0) | 41 (15.4) [f] | 299 (39.9) | < .001 |
| **Female** | | | | | | |
| Number | 3946 | 411 | 664 | 893 | 1978 | - |
| Age(year) | 42.3 ± 13.1 | 30.7 ± 10.9 | 35.0 ± 10.3[e] | 40.1 ± 11.2 | 48.2 ± 12.0 | < .001 |
| Weight(kg) | 71.4 ± 11.7 | 53.7 ± 5.6 | 70.0 ± 8.2[e] | 71.2 ± 8.9 | 75.6 ± 11.1 | |
| BMI (kg/m$^2$) | 29.2 ± 4.5 | 21.2 ± 1.6 | 28.3 ± 2.9[e] | 29.1 ± 3.3 | 31.2 ± 4.0 | < .001 |
| WC (cm) | 91.5 ± 12.0 | 72.6 ± 6.5 | 85.8 ± 8.6[e] | 89.4 ± 9.1 | 98.2 ± 9.1 | < .001 |
| Smoker (%) | 113 (2.9) | 8 (2) | 21 (3.2) | 24 (2.7) | 60 (3.0) | .62 |
| Education (%) | | | | | | |
| Less than diploma and diploma | 3589 (91.0) | 311 (75.7) | 577 (86.9) | 812 (90.9) | 1889 (95.6) | < .001 |
| Higher than diploma | 355 (9.0) | 100 (24.3) | 87 (13.1) | 81 (9.1) | 87 (4.4) | < .001 |
| Physical activity (%) | | | | | | |
| Low | 2627 (66.8) | 253 (62.0) | 422 (63.8) | 517 (64.2) | 1381 (70.0) | < .001 |
| High | 1304 (33.2) | 155 (38.0) | 239 (36.2) | 318 (35.8) | 592 (30.0) | < .001 |
| Family history of premature CAD (%) | 681 (17.3) | 53 (12.9) | 98 (14.8) | 133 (14.9) | 397 (20.1) | < .001 |
| SBP (mmHg) | 120.1 ± 19.7 | 105.7 ± 11.9 | 109.6 ± 12.0[e] | 114.3 ± 14.0 | 129.3 ± 20.7 | < .001 |
| DBP (mmHg) | 78.5 ± 10.8 | 69.8 ± 8.7 | 73.4 ± 8.0[e] | 75.5 ± 8.4 | 83.4 ± 0.5 | < .001 |
| Hypertension[b] (%) | 2958 (75.2) | 400 (97.7) [e] | 641 (96.5) | 809 (90.7) | 1108 (56.3) | < .001 |
| FBS (mg/dL) | 99.4 ± 35.0 | 85.5 ± 19.8 | 86.3 ± 8.7 | 89.4 ± 15.4 | 112.7 ± 44.9 | < .001 |
| 2-hBG | 123.5 ± 54.8 | 94.6 ± 20.8 | 101.3 ± 24.3[e] | 109.3 ± 28.1 | 145.4 ± 68.1 | < .001 |

*(Continued)*

**Table 1.** (Continued)

| | Total | Persistent MHNW | Persistent MHO | Transition from MHO to MUO | MUO | P-value |
|---|---|---|---|---|---|---|
| Diabetes (%) | 495 (13.0) | 2 (0.5) | 4 (0.6) | 17 (1.9) | 472 (24.5) | < .001 |
| HDL cholesterol (mg/dL) | 43.6 ± 10.7 | 47.1 ± 10.8 | 48.6 ± 11.9e | 44.7 ± 11.0 | 40.6 ± 9.1 | < .001 |
| Low HDL cholestrole c (%) | 3047 (77.4) | 272 (66.5) | 387 (58.4) e | 631 (70.7) | 1757 (89.1) | < .001 |
| Total cholesterol (mg/dL) | 214.0 ± 48.2 | 175.1 ± 36.0 | 193.9 ± 36.3e | 207.1 ± 42.0 | 232.0 ± 48.3 | < .001 |
| Triglycerides(mg/dL) | 149 (101–212) | 83 (64–126) | 97 (76–126) e | 125 (97–155) | 199 (158–262) | < .001 |
| Triglycerides ≥ 150 mg/dl | 1967 (49.9) | 30 (7.3) | 77 (11.6) | 242 (27.1) | 1618 (82.0) | < .001 |
| Insulin d(Mu/L) | 8.6 (6.2–11.7) | 6.1 (4.4–8.3) | 7.7 (5.2–10.5)e | 8.2 (6.211.2) | 9.7 (7.2–13.1) | < .001 |
| HOMA-IR d(mole ×mU/l2) | 1.9 (1.3–2.8) | 1.2 (.9–1.7) | 1.6 (1.0–2.2) e | 1.8 (1.3–2.4) | 2.4 (1.7–3.6) | < .001 |
| IR d (%) | 620 (29.6) | 12 (5.7) | 60 (15.7) e | 128 (21.8) | 420 (45.9) | < .001 |

MHNW, metabolically healthy normal weight; MHO, metabolically healthy overweight/obese; MHO, metabolically healthy overweight/obese; MUO, Metabolically unhealthy overweight/obese; BMI, body mass index; WC, waist circumference; CAD, coronary artery disease; SBP, systolic blood pressure; DBP, diastolic blood pressure; HDL-C, high-density lipoprotein cholesterol; FBS, fasting blood sugar; 2-h BG, 2-h blood glucose; HOMA-IR, homeostatic model assessment-insulin resistance; IR, insulin resistance. Values are expressed as mean (SD), median (IQR 25–75), or percentages.

a Metabolic health defined as ≤2 components of metabolic syndrome according to joint interim statement (JIS) definition.

b Hypertension defined as SBP ≥135mmHg and/or DBP ≥80mmHg and/or antihypertensive drug use.

c Low HDL-C defined as HDL-C <40/50 mg/dl for men/women.

d Measured or calculated in 3946 Subjects; IR defined as HOMA-IR 2.6 mole mU/l2.

e Comparison between persistent MHO and transition from MHO to MUO groups, p < 0.001

f Comparison between Males and Females who transition from MHO to MUO, p < 0.001

model [HR = 1.17 (.77–1.78)]; however, after adjustment for age, and also, in the fully adjusted model, CVD risk became statistically significant [HR = 1.62 (1.06–2.46), HR = 1.55 (1.02–2.37)]. Moreover, a sensitivity analysis was conducted after excluding individuals with known Diabetes. Nevertheless, the results were similar to our main analysis. Furthermore, we did the analysis after entering the interaction of Diabetes and obesity phenotypes into the model; leading to the same results.

## Discussion

Findings of this population-based cohort study showed that over 15.9-years of follow-up, transition from MHO to MUO phenotype increased the risk of CVD incidence in male subjects but not in females. On the other hand, in both sex groups, persistent MHO phenotype was not associated with higher risk of CVD incidence. According to our findings, the MHO phenotype is a heterogeneous status which is caused by transition to MUO phenotype over time.

MHO phenotype and its associated CVD risk has been a challenging subject. several studies found that MHO phenotype is not completely a benign condition [9, 13, 14, 24, 25], though few studies have reported otherwise [6, 8]. The dynamic feature of MHO and its transition to the MUO phenotype could be a reason for these inconsistent results. Recently, another a few studies also considered the transitional feature of MHO phenotype and its associated CVD risk [14, 15, 26].

Studies regarding the transition of MHO status to MUO and its associated CVD outcome have shown different results. Notably, these studies differ in length of follow-up, definition of healthy status, sample size, adjustments and outcome verification. In the present 15.9-year follow-up study, it was demonstrated that MHO has a dynamic feature that divides this phenotype into persistent and transitional groups. Moreover, we found that unlike the persistent MHO status, transitional counterparts had a higher risk for CVD outcomes only in male participants.

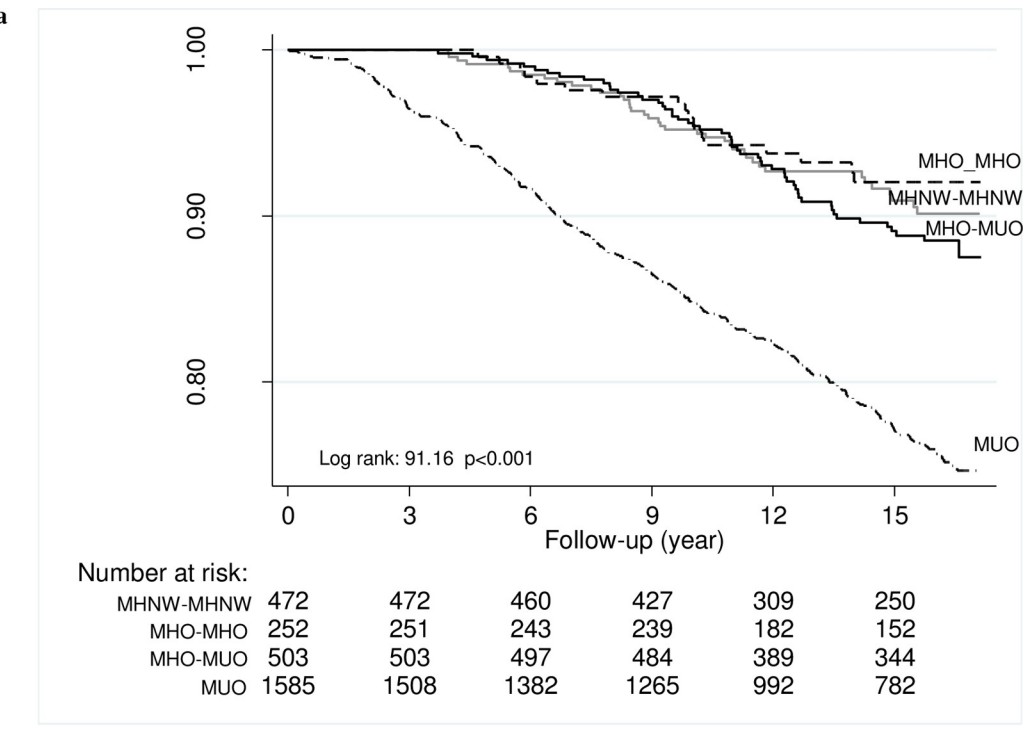

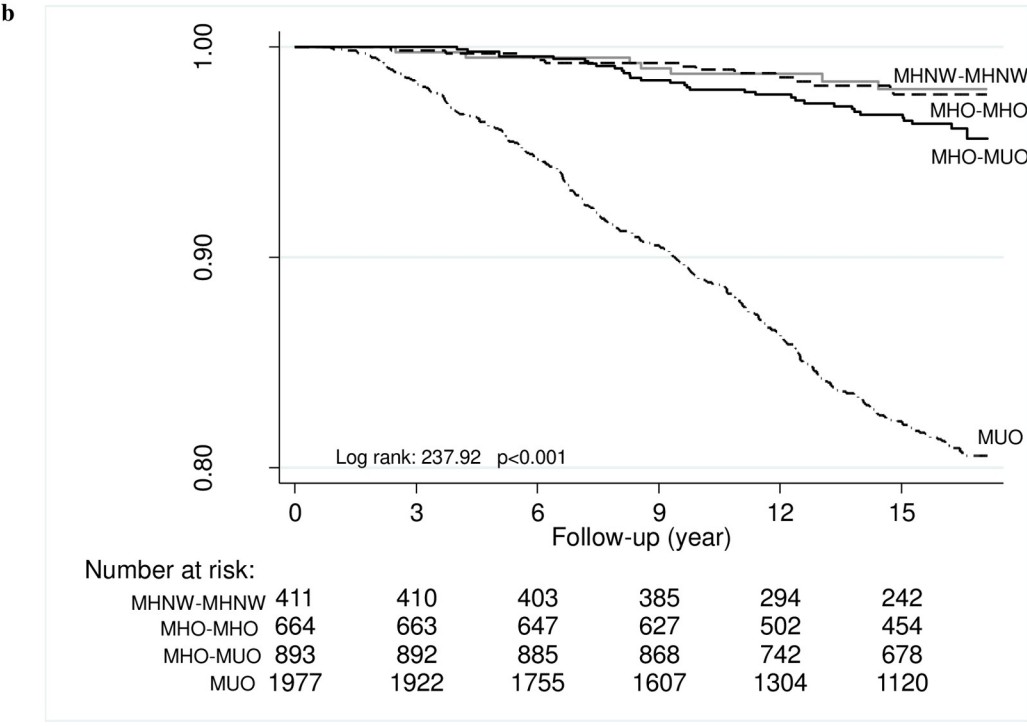

**Fig 2. Kaplan-Meier curves for cumulative survival free from cardiovascular events as a function of obesity phenotypes according to body mass index and metabolic health in each obesity phenotype. A**-Male, **B**-Female.

**Table 2. Hazard ratios (HRs) for incident cardiovascular disease in 6758 study participants according to obesity phenotypes at baseline and through 15-years of follow-up.**

| | Male | | | | Female | | | |
|---|---|---|---|---|---|---|---|---|
| | **MHNW** n = 472 | **Persistent MHO** n = 252 | **MHO to MUO** n = 503 | **MUO** n = 1585 | **MHNW** n = 411 | **Persistent MHO** n = 664 | **MHO to MUO** n = 893 | **MUO** n = 1978 |
| No of person-years | 6470 | 3566 | 7361 | 20135 | 5836 | 9695 | 13526 | 26026 |
| No of incident CVD | 39 | 18 | 53 | 340 | 7 | 13 | 32 | 326 |
| Incidence rate (per 1,000 person-years) | 6 | 5 | 7 | 17 | 1 | 1 | 2 | 12 |
| HR[a] (95% CI) | 1 | .83 (0.47–1.45) | 1.17 (0.77–1.78) | **2.83 (2.03–3.94)** | 1 | 1.11 (0.44–2.78) | 1.94 (0.86–4.41) | **10.57 (5.00–22.36)** |
| HR[b] (95% CI) | 1 | 1.28 (0.73–2.25) | **1.62 (1.06–2.46)** | **2.83 (2.03–3.95)** | 1 | 0.91 (0.36–2.28) | 1.09 (0.48–2.48) | **3.50 (1.64–7.46)** |
| HR[c] (95% CI) | 1 | 1.37 (0.78–2.41) | **1.55 (1.02–2.37)** | **2.64 (1.89–3.70)** | 1 | 0.85 (0.34–2.15) | 0.93 (0.41–2.12) | **2.65 (1.24–5.68)** |

MHNW, metabolically healthy normal weight; MHO, metabolically healthy overweight/obese; MUO, metabolically unhealthy overweight/obese; CVD, cardiovascular disease

[a] unadjusted model

[b] adjusted for age

[c] adjustment for age, physical activity, total cholesterol, education, smoking, family history CVD

Similar to our study, Mongraw-Chaffin et al. [26], during a 12 year follow-up, found that MHO is not a stable condition and almost one-half of those with MHO phenotype at baseline, developed metabolic abnormalities. Transition from MHO to MUO phenotype was associated with a higher risk for CVD, although it was lower than the risk of those with MUO at baseline. On the other hand, the persistent MHO phenotype was not associated with higher risk of CVD. Noticeably, the absolute risk of CVD incidence in persistent MHO and transitional MHO was 0.07 and 0.1 in males. Therefore, transition increased the risk of CVD incidence by 3%. In other words, by transition of 33 cases from MHO to MUO status, one case of CVD would be developed, which is of great importance taking into consideration the high probability of transition occurrence.

In contrast to our study, Eckel et al. [14], in a 30 year longitudinal follow-up (the Nurses' Health Study), reported that females with MHO phenotype at baseline, even without transition to MUO phenotype, were at higher risk for CVD. This finding can be explained by the long follow-up period in this study, which gives enough time for the MHO phenotype to reveal its effect on CVD outcome. However, it is noteworthy that metabolic health in the aforementioned study was defined as having none of the metabolic disorders including hypertension, diabetes and hypercholesterolemia. Also, BMI, beside all metabolic disorders, was assessed based on self-reported questionnaires. Moreover, Matina Kouvari et al. [15] reported that persistent MHO compare to MHNW phenotype had a higher chance of presenting CVD events during 10 years of follow-up. Additionally, transition from MHO to MUO phenotype significantly increased the risk of CVD. This endorses the hypothesis that MHO phenotype is a dynamic status and also not a completely benign condition. In this study, a strict definition of healthy metabolic status was used. MHO was defined as having none of the metabolic syndrome criteria with an exception of waist circumference, since most obese individuals have waist circumferences above the normal range [9].

Sex impact on CVD is a well-established concept [27, 28]. Several studies investigated metabolic disorders and cardiovascular diseases, with females generally having a more beneficial metabolic profile and lower cardiovascular diseases [29–31]. Consistently, we found that compared to males, females had a lower rate of CVD incidence, and also, exhibited a healthier

metabolic profile at baseline. Moreover, transitional MHO females had a healthier metabolic profile than their male counterparts at baseline. Of note, in males but not females, main different between Persistent MHO and people who transition is in Diabetes, FBG and IR, which could be the reason why only in males transitional MHO had an association with CVD incidence. However, after conducting a sensitivity analysis excluding the individuals with known diabetes, the same results have been emerged. All taken together, we believe that the more favorable metabolic profile at baseline, lower CVD incidence rate in females, sexual dimorphism in body fat composition [32] and more importantly, a healthier metabolic profile in transitional MHO females somehow explain the benign nature of transitional MHO phenotype. However, for further clarifications, special attention must be paid towards sex differences in genetic, nutritional and socioeconomic factors as well [33–35].

This study has some limitations. First of all, information on socioeconomic status and nutrition of subjects was not available. Moreover, in the first phase of TLGS physical activity was recorded using the Lipid Research Clinic questionnaire in the first phase of TLGS, which has not been validated in Iran. Secondly, defining the MHO phenotype as strict as just having 1 or lack of any metabolic criteria was not feasible, due to the scarce number of outcomes in this group. Finally, low number of events in females in transitional MHO group could be a reason that an association with CVD incidence was not found in this group. On the other hand, the current study has some strength too. First of all, to the best of our knowledge, this was a unique dataset in an underrepresented group. Secondly, the long-term follow-up allowed us to shed light upon the heterogeneity of MHO phenotype and its dynamic feature. Moreover, actually measures of variables and outcomes were used rather than self-reported data.

In conclusion, this study revealed that obesity itself, without causing a metabolically unhealthy status, doesn't increase the risk of CVD in both sex groups during 15.9 year follow up. On the other hand, transition to metabolically unhealthy status was associated with higher risk of CVD only in males and not in females. Therefore, one of the reasons for the heterogeneity in MHO status is its transition to MUO phenotype. However, further studies with larger sample sizes and longer follow-ups are needed to investigate the underlying factors for MHO heterogeneity. Additionally, Studies inquiring the persistent MHO status and the features protecting them from transition to MUO status are conducive as well.

## Acknowledgments

We would like to acknowledge the staff and participants of the TLGS study for their important contribution, and also Dr. Forough Ghanbari for critical editing of English grammar and syntax of the manuscript.

## Author Contributions

**Conceptualization:** Farhad Hosseinpanah.

**Formal analysis:** Maryam Mahdavi.

**Investigation:** Erfan Tasdighi.

**Methodology:** Farhad Hosseinpanah, Erfan Tasdighi.

**Software:** Maryam Mahdavi.

**Supervision:** Arash Ghanbarian, Majid Valizadeh, Fereidoun Azizi.

**Writing – original draft:** Erfan Tasdighi.

**Writing – review & editing:** Farhad Hosseinpanah, Erfan Tasdighi, Maryam Barzin.

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
