## [Decision Letter · Decision Letter 0]

22 Jul 2020

PONE-D-20-19533

The association between transition from metabolically healthy obesity to metabolic syndrome, and incidence of cardiovascular disease: Tehran lipid and glucose study,

PLOS ONE

Dear Dr. Hosseinpanah,

Thank you for submitting your manuscript to PLOS ONE. After careful consideration, we feel that it has merit but does not fully meet PLOS ONE’s publication criteria as it currently stands. Therefore, we invite you to submit a revised version of the manuscript that addresses the points raised during the review process.

We look forward to receiving your revised manuscript.

Kind regards,

Antonio Palazón-Bru, PhD

Academic Editor

PLOS ONE

Journal Requirements:

2. Please address the following:

- Please include additional information regarding the survey or questionnaire used in the study and ensure that you have provided sufficient details that others could replicate the analyses.

For instance, if you developed a questionnaire as part of this study and it is not under a copyright more restrictive than CC-BY, please include a copy, in both the original language and English, as Supporting Information.

- Please refer to any post-hoc corrections to correct for multiple comparisons during your statistical analyses/sample size calculations performed prior to participant recruitment.

If these were not performed please justify the reasons.

Please refer to our statistical reporting guidelines for assistance (https://journals.plos.org/plosone/s/submission-guidelines.#loc-statistical-reporting).

4. Your ethics statement must appear in the Methods section of your manuscript.

If your ethics statement is written in any section besides the Methods, please move it to the Methods section and delete it from any other section.

Please also ensure that your ethics statement is included in your manuscript, as the ethics section of your online submission will not be published alongside your manuscript.

Reviewers' comments:

Reviewer's Responses to Questions

**Comments to the Author**

1. Is the manuscript technically sound, and do the data support the conclusions?

Reviewer #1: Yes

Reviewer #2: Yes

2. Has the statistical analysis been performed appropriately and rigorously? 

Reviewer #1: Yes

Reviewer #2: Yes

3. Have the authors made all data underlying the findings in their manuscript fully available?

Reviewer #1: No

Reviewer #2: No

4. Is the manuscript presented in an intelligible fashion and written in standard English?

Reviewer #1: Yes

Reviewer #2: Yes

5. Review Comments to the Author

Reviewer #1: The manuscript from Hosseinpanah et al presents the results of a prospective study on the association cardiovascular diseases (CVD) and the transition from a metabolically healthy obese phenotype to metabolic syndrome. Authors report that an increased occurrence of CVD in those individuals transitioning from a metabolically healthy obese phenotype to metabolic syndrome, compared with those who remain in a metabolically healthy obese phenotype and non-obese controls.

The manuscript is very well written and the hypothesis is scientifically sound. Results are based on a large patient cohort, and the methods are appropriate. The study design (in terms of group's compositions) is equilibrated, thus minimizing the impact of confounding effects. Overall, I consider that this manuscript will be of interest for a large audience.

My only concern is that one of the key findings reported in the manuscript is the CVD incidence in the follow up is higher in males than in females (12 and 7 1000 persons per year, respectively). However, Table 1 shows that whereas education (as a proxy of socioeconomic status) and physical activity percentages do not show statistically significant differences among the groups in males (p>0.05), those parameters are statistically different in females (p < 0.001). This is a potential source of confounders. Authors should acknowledge this and detail the actions for minimizing the impact of such differences.

Reviewer #2: What was your apriori hypothesis?

"Mean" follow up of 15 years?

What would have your results been if you had not combined overweight and obese (six groups)? Are results similar if overweight is distinguished from obese?

Perhaps your groups should be labeled overweight/obese to be much clearer/precise?

Please clarify which predictors, if any, are time-varying, particularly cross-categorizations of overweight/obesity and metabolically healthy. If time-invariant, please specify whether they were taken from baseline.

Why weren't the models adjusted for prevalent diabetes?

Please provide additional pairwise comparisons from the Cox model.

6. PLOS authors have the option to publish the peer review history of their article (what does this mean?). If published, this will include your full peer review and any attached files.

Reviewer #1: No

Reviewer #2: No

---

## [Author Response · Author response to Decision Letter 0]

14 Aug 2020

Dear Editor-in-Chief,

We would like to thank you for your thoughtful comments. We have taken this opportunity to improve our manuscript and taking into account all the points raised by the reviewer on different aspects of our report. 

We have taken each critique and comments of reviewers’ very seriously. Herewith, is a revised version of the manuscript with a point by point response to each comment and all changes have been highlighted. We are looking forward to hearing from you at your earliest convenience. We hope that you find our explanations below and changes in the manuscript fulfilling.

Yours Sincerely,

Farhad Hosseinpanah, MD

Response to Reviewer #1:

Comment #1: My only concern is that one of the key findings reported in the manuscript is the CVD incidence in the follow up is higher in males than in females (12 and 7 1000 persons per year, respectively). However, Table 1 shows that whereas education (as a proxy of socioeconomic status) and physical activity percentages do not show statistically significant differences among the groups in males (p>0.05), those parameters are statistically different in females (p < 0.001). This is a potential source of confounders. Authors should acknowledge this and detail the actions for minimizing the impact of such differences.

Response: Thank you for your precisive review of our paper. Based on the ANOVA analysis the education and physical activity were significantly different across phenotypes among female group. However, pairwise comparisons between persistent MHO and transitional MHO with the reference group (MHNW) did not reveal any difference in this regard. Moreover, between-group analysis showed that there were no differences between male and females in persistent and transitional MHO regarding education status and physical activity. Nevertheless, considering that education status and physical activity level could be possible confounders the multivariate analysis was conducted with adjustment for both education and physical activity. 

The within-group differences regarding physical activity and education status in females were mentioned in the results.

Results section (Page 10, Line 226-231)

“In contrast to male participants, there was a significant difference in females regarding education status and physical activity level; however, pairwise comparisons between persistent MHO and transitional MHO with the reference group (MHNW) did not reveal any difference in this regard. Moreover, between-group analysis showed that there were no differences between male and females in persistent and transitional MHO regarding education status and physical activity.”

Response to Reviewer #2:

Comment #1: What was you’re a priori hypothesis?

Response: We are sincerely grateful for your consideration and guidance which helped us to improved our paper and make it clearer.

We had two H0 hypothesis:

1. “Persistent MHO status is not associated with higher risk of CVD incidence, in comparison with MHNW”

2. “Transition from MHO to MUO status is not associated with higher risk of CVD incidence, in comparison with MHNW”

Comment #2."Mean" follow up of 15 years?

Response: Agreed and corrected. the median (IQ25-75) follow-up was 15.9 (11.6-16.4) and median follow up was mentioned throughout the entire paper.

Comment #3. What would have your results been if you had not combined overweight and obese (six groups)? Are results similar if overweight is distinguished from obese?

Response: Since the number of participants and the outcome incidence rate of outcome would be much less if we defined obesity as BMI≥30 kg/m2, we tried to expand our study population by combining overweight and obesity (BMI≥ 25 kg/m2).In fact the power of our analysis was inadequate to define obesity based on BMI≥30kg/m2 . Moreover, it is important to note that, majority of Iranian population who have excess weight are overweight not obese. Taken all together we decided to combine these two groups (obese and overweight) and consider them as one. Similar to us ,previous studies adopted the same approach as well, defining obesity for MHO individuals as BMI≥ 25 kg/m2.(1),(2). 

Comment #4. Perhaps your groups should be labeled overweight/obese to be much clearer/precise?

Response: Agreed and corrected.

Material and method section (Page 7, Line 167-173)

“Overweight/obesity was defined as BMI ≥ 25 kg/m2. According to BMI categories and metabolic status, participants were divided into 4 groups: (1) metabolically healthy normal weight (MHNW) defined as BMI<25kg/m2 and healthy metabolic status; (2) metabolically healthy overweight/obese (MHO) defined as BMI≥25kg/m2 and healthy metabolic status; (3) metabolically unhealthy normal weight (MUNW) defined as BMI<25kg/m2 and unhealthy metabolic status; (4) metabolically unhealthy overweight/obese (MUO) defined as BMI≥25kg/m2 and unhealthy metabolic status”

Comment #5. Please clarify which predictors, if any, are time-varying, particularly cross-categorizations of overweight/obesity and metabolically healthy. If time-invariant, please specify whether they were taken from baseline.

Response: Agreed and corrected. In this study, Cox proportional hazards model was used, not the Cox time-dependent model. In fact, transition from metabolically healthy obesity to metabolic syndrome was time-varying until incidence of cardiovascular disease or till the end of follow-up. This variable, due to the nature of its definition, had to change over time. Therefore, BMI and all metabolic criteria (FBS, 2hpG, TG, HDL-C), SBP, DBP and waist circumference were measure at baseline and at each phase of TLGS which were three years apart until the occurrence of the outcome or till the end of follow-up. 

statistical analysis section (Page 8, Line 187-191)

“Transition from metabolically healthy obesity to metabolic syndrome was time-varying until incidence of cardiovascular disease or till the end of follow-up. This variable, due to the nature of its definition, had to change over time. Therefore, BMI and all metabolic criteria (FBS, 2hpG, TG, HDL-C), SBP, DBP and waist circumference were measure at baseline and at each phase of TLGS which were three years apart until the occurrence of the outcome or till the end of follow-up.”

Comment #6. Why weren't the models adjusted for prevalent diabetes?

Response: Agreed and corrected. it is important to note that the majority (65.5%) of the participants with diabetes were newly diagnosed and 34.5% had known diabetes. We have had conducted a sensitivity analysis after excluding individuals with known Diabetes. Nevertheless, the results were similar to our main analysis. Moreover, as you have suggested we did the analysis after entering the interaction of Diabetes and obesity phenotypes into the model; leading to the same results. We added this point to the results.

Results section (Page 11, Line 247-251)

“Moreover, a sensitivity analysis was conducted after excluding individuals with known Diabetes. Nevertheless, the results were similar to our main analysis. Furthermore, we did the analysis after entering the interaction of Diabetes and obesity phenotypes into the model; leading to the same results.”

Table. Analysis for CVD incidence after adjusting the interaction of diabetes and obesity phenotype into the model.

 Male Female

 MHNW Persistent MHO MHO to MUO MUO MHNW Persistent MHO MHO to MUO MUO

HRa (95% CI) 1 .83 (0.46-1.48) 1.12 (0.72-1.73) 2.18 (1.53-3.10) 1 1.36 (0.48-3.88) 2.11 (0.81-5.50) 9.58 (3.93-23.30)

HRb (95% CI) 1 1.23 (0.69-2.20) 1.52 (0.99-2.37) 2.30 (1.63-3.32) 1 1.14 (0.40-3.23) 1.27 (0.49-3.33) 3.82 (1.56-9.35)

HRc (95% CI) 1 1.330 (0.74-2.38) 1.49 (0.97-2.33) 2.20 (1.54-5.11) 1 1.05 (0.37-3.00) 1.09 (0.41-2.84) 2.89 (1.17-7.12)

a unadjusted model

b adjusted for age 

c adjustment for age, physical activity, total cholesterol, education, smoking, family history CVD 

Comment #7. Please provide additional pairwise comparisons from the Cox model.

 Response: As we have mentioned before, a prior hypothesis was the comparison of persistent MHO and transitional MHO with the MHNW group which was presented in table 2. Nevertheless, as you have requested a pairwise comparison between transitional MHO and persistent MHO (as the reference group) was conducted. As it can be expected, due to the few number of cases in the persistent MHO group there was inadequate power regarding these pairwise comparisons. 

Table. Pairwise analysis for CVD incidence 

 Male Female

 Persistent MHO MHO to MUO Persistent MHO MHO to MUO

HRa (95% CI) 1 1.40(.82-2.40) 1 1.73 (.91-2.40)

HRb (95% CI) 1 1.22 (.72-2.09) 1 1.05 (.54-2.00)

HRc (95% CI) 1 1.10 (.64-1.89) 1 1.05 (.54-2.02)

a unadjusted model

b adjusted for age 

c adjustment for age, physical activity, total cholesterol, education, smoking, family history CVD

---

## [Decision Letter · Decision Letter 1]

1 Sep 2020

The association between transition from metabolically healthy obesity to metabolic syndrome, and incidence of cardiovascular disease: Tehran lipid and glucose study,

PONE-D-20-19533R1

Dear Dr. Hosseinpanah,

We’re pleased to inform you that your manuscript has been judged scientifically suitable for publication and will be formally accepted for publication once it meets all outstanding technical requirements.

Kind regards,

Antonio Palazón-Bru, PhD

Academic Editor

PLOS ONE

Additional Editor Comments (optional):

Reviewers' comments:

Reviewer's Responses to Questions

**Comments to the Author**

1. If the authors have adequately addressed your comments raised in a previous round of review and you feel that this manuscript is now acceptable for publication, you may indicate that here to bypass the “Comments to the Author” section, enter your conflict of interest statement in the “Confidential to Editor” section, and submit your "Accept" recommendation.

Reviewer #1: All comments have been addressed

Reviewer #2: All comments have been addressed

2. Is the manuscript technically sound, and do the data support the conclusions?

Reviewer #1: Yes

Reviewer #2: Yes

3. Has the statistical analysis been performed appropriately and rigorously? 

Reviewer #1: Yes

Reviewer #2: Yes

4. Have the authors made all data underlying the findings in their manuscript fully available?

Reviewer #1: Yes

Reviewer #2: No

5. Is the manuscript presented in an intelligible fashion and written in standard English?

Reviewer #1: Yes

Reviewer #2: Yes

6. Review Comments to the Author

Reviewer #1: (No Response)

Reviewer #2: (No Response)

7. PLOS authors have the option to publish the peer review history of their article (what does this mean?). If published, this will include your full peer review and any attached files.

Reviewer #1: No

Reviewer #2: No

---

## [Editor Report · Acceptance letter]

3 Sep 2020

PONE-D-20-19533R1 

The association between transition from metabolically healthy obesity to metabolic syndrome, and incidence of cardiovascular disease: Tehran lipid and glucose study, 

Dear Dr. Hosseinpanah:

I'm pleased to inform you that your manuscript has been deemed suitable for publication in PLOS ONE. Congratulations! Your manuscript is now with our production department. 

Kind regards, 

on behalf of

Dr. Antonio Palazón-Bru 

Academic Editor

PLOS ONE